# Seismic Performance of Corroded ECC-GFRP Spiral-Confined Reinforced-Concrete Column

**DOI:** 10.3390/polym16152110

**Published:** 2024-07-24

**Authors:** Xu Long, Zehong Chen, Pengda Li

**Affiliations:** Guangdong Provincial Key Laboratory of Durability for Marine Civil Engineering, Shenzhen University, Shenzhen 518060, China; longx@szu.edu.cn (X.L.);

**Keywords:** engineered cementitious composites (ECCs), GFRP spiral, RC column, durability, seismic behavior, FE analysis

## Abstract

Preventing corrosion in the steel reinforcement of concrete structures is crucial for maintaining structural integrity and load-bearing capacity as it directly impacts the safety and lifespan of concrete structures. By preventing rebar corrosion, the durability and seismic performance of the structures can be significantly enhanced. This study investigates the hysteresis behavior of both corroded and non-corroded engineered cementitious composite (ECC)-glass-fiber-reinforced polymer (GFRP) spiral-confined reinforced-concrete (RC) columns. Employing experimental methods and finite element analysis, this research explores key seismic parameters such as crack patterns, failure modes, hysteretic responses, load-bearing capacities, ductility, stiffness degradation, and energy dissipation. The results demonstrate that ECC-GFRP spiral-confined RC columns, compared to traditional RC columns, show reduced corrosion rates, smaller crack widths, and fewer corrosion products, indicating superior crack control and corrosion resistance. Hysteresis tests revealed that ECC-GFRP columns, at a 20% target corrosion rate, exhibit an enhanced load-bearing capacity, ductility, and energy dissipation, suggesting improved durability and seismic resilience. Parametric and sensitivity analyses confirm the finite element model’s accuracy and highlight the significant influence of concrete compressive strength on load-bearing capacity. The findings suggest that ECC-GFRP spiral-confined RC columns offer promising applications in coastal and seismic-prone regions, enhancing corrosion resistance and mechanical properties, thus potentially reducing formwork costs and improving construction quality and efficiency.

## 1. Introduction

Coastal infrastructure frequently suffers from erosion due to sea currents, waves, marine erosion, and tides. In such corrosive environments, the intrusion of chloride ions into concrete structures leads to rebar corrosion, resulting in structural performance degradation, which severely impacts structural safety and incurs substantial economic losses [1,2,3]. Corrosion of the rebar results in a reduction in the effective cross-sectional area, leading to decreased strength and ductility. Additionally, corrosion products on the rebar surface impair the bond performance with concrete, diminishing both the load-bearing capacity and ductility of the structure [4,5,6], thereby reducing its seismic performance. For coastal reinforced-concrete bridge piers and frame structures in seismically active regions, traditional RC column seismic design enhances energy dissipation capacity and prevents brittle failure by increasing the space between stirrups in the plastic hinge region at the column base. However, in corrosive environments, stirrups, being on the outermost position of the reinforcement framework, corrode more severely and rapidly than longitudinal reinforcement [7]. The corrosion of stirrups not only accelerates the corrosion of the main reinforcement, but also leads to the loss of lateral confinement, changing the failure mode from ductile to shear brittle failure [7]. Therefore, ensuring the seismic performance of RC columns under the combined effects of seismic loads and corrosive environments presents a greater challenge.

Current measures to inhibit rebar corrosion primarily aim to block the transmission of corrosive agents. For instance, increasing the thickness of the concrete cover can prolong the time it takes for corrosive agents to reach the rebar surface [8]. However, due to the brittle nature of concrete, it is prone to cracking characteristics, and the volumetric expansion that occurs due to rebar corrosion can cause the concrete cover to crack and spall, reducing the structural mechanical performance and further accelerating rebar corrosion. Additionally, applying coatings such as corrosion inhibitors or epoxy resin to the rebar surface to prevent the ingress of corrosive agents is another common method to extend the lifespan of reinforced-concrete structures [9]. However, these coatings compromise the bond performance between the rebar and concrete, hindering their cooperative behavior [9,10]. In recent years, with advancements in material science and significant reductions in the production costs of new materials, replacing steel rebar with corrosion-resistant fiber-reinforced polymer (FRP) bars has become an effective measure to prevent structural degradation that occurs due to rebar corrosion [11,12]. FRP materials also have excellent tensile properties. For example, carbon fiber (CFRP) has a tensile strength that can reach 4000 MPa. These performance advantages make FRP materials widely used in improving the performance of concrete structures or for structural rehabilitation [13,14,15,16,17]. However, the compressive performance of FRP materials is significantly inferior to their tensile performance. As longitudinal reinforcement, FRP bars have a compressive strength that is only 30–50% of their tensile strength [18]. Additionally, the inherently brittle failure mode of FRP materials also limits their application in compressive members. Therefore, balancing the advantages and disadvantages of material applications and different combinations is crucial. Understanding the performance characteristics of various materials and their reasonable combination and application is key to enhancing both the corrosion resistance and seismic performance of RC columns.

This study proposes a novel hybrid, confined reinforced-concrete (RC) column structure, wherein FRP spirals replace the steel stirrups embedded in engineered cementitious composites (ECCs). This design involves the FRP and ECCs as both a confining element for reinforced-concrete columns and a corrosion-resistant protective layer, enhancing the structural corrosion resistance and mechanical performance. This structural form leverages the multiple fine cracking characteristics of ECCs to act as a protective layer, blocking the intrusion of corrosive agents [19,20]. Additionally, the ultra-high tensile deformation capacity of ECCs (over 8%) [21,22,23] can also improve the seismic ductility and energy dissipation capacity of reinforced-concrete columns [24,25,26,27,28,29]. However, the production cost of ECCs is still relatively high compared to ordinary concrete, making large-scale use economically unfeasible. Previous studies have proposed ECC-precast jacket-reinforced-concrete-composite columns, where the ECCs constitute 56–64% of the total cross-sectional area, and yet the seismic performance is comparable to that of full ECC reinforced-concrete columns [25,30]. Cai et al. [31] suggested replacing concrete with ECCs for externally confined steel-tube concrete columns. However, the hysteresis test results indicated that the ECCs’ contribution to the load-bearing capacity of composite columns as a confinement material was limited. Therefore, combining the excellent corrosion resistance and high tensile strength of FRPs with ECCs as the protective layer for core concrete is a potential method for simultaneously enhancing the durability and seismic performance of RC columns in coastal environments. Using FRP stirrups and ECCs ensures corrosion resistance and improves confinement while reducing ECC usage and enhancing the load-bearing capacity and ductility of the core concrete.

To thoroughly enhance the corrosion resistance of concrete columns, previous researchers, such as Afifi et al. [32] and Hadi et al. [33], have tried to completely replace steel rebar with glass FRP (GFRP) bars in reinforced-concrete columns, and they have conducted axial and eccentric compression tests. Their results indicate that the axial load-bearing capacity and ductility of GFRP-RC columns are similar or superior to those of RC columns. However, Abdallah and El-Salakawy [34] conducted hysteresis tests on five GFRP-RC columns and one steel-reinforced-concrete column. The test results showed that the GFRP-RC columns exhibited load-bearing capacities similar to those of steel-reinforced-concrete columns. However, due to the linear elastic nature of GFRP longitudinal reinforcement, the energy dissipation of GFRP-RC columns, primarily arising from the plastic deformation of longitudinal reinforcement, was only half that of steel-reinforced-concrete columns. Additionally, Tavassoli and Sheikh [35] found that GFRP longitudinal rebar–steel stirrup concrete columns have lower flexural strength under seismic loads compared to steel-reinforced-concrete columns. These studies further illustrate that while replacing longitudinal reinforcement with FRP bars enhances corrosion resistance, it compromises seismic performance. Therefore, replacing rebar with FRP bars entirely may not be a perfect option. Pantelides et al. [36] only replaced confinement steel stirrups with FRP spirals and compared the axial mechanical performance of GFRP spiral-steel longitudinal rebar concrete columns and traditional steel-reinforced-concrete columns in corrosive environments. Concrete columns with composite GFRP spirals exhibited lower rebar corrosion rates and superior axial load-bearing capacity and ductility. This application further validated that the proposed ECC-GFRP spiral-confined RC column can ensure corrosion resistance while using steel rebars as longitudinal reinforcement to maintain energy dissipation capacity under cyclic loading. However, the current research on such composite structural columns is limited to the axial mechanical performance of short columns, and research on the seismic performance of these structural columns in corrosive environments remains unexplored.

This study experimentally investigated the seismic performance of ECC-GFRP spiral-confined RC columns under severe corrosive conditions, examining their failure modes, load-bearing capacity, ductility, stiffness degradation, and energy dissipation capabilities. In addition, based on the structural form of these novel ECC-GFRP spiral-confined RC columns, a finite element model was established and validated for parameter study. Through the finite element parameter studies and sensitivity analyses, the effects of design parameters such as longitudinal reinforcement ratio, aspect ratio, concrete compressive strength, and ECC compressive strength on the seismic performance of ECC-GFRP spiral-confined RC columns were further investigated.

## 2. Experimental Program

### 2.1. Specimen Design

This study primarily investigated and compared the corrosion resistance of ECC-GFRP spiral-confined RC columns with traditional RC columns, and it examined their mechanical bearing capacity and seismic performance post-corrosion. In this experimental study, two traditional RC columns (C0-RC and C20-RC) and two ECC-GFRP spiral-confined RC columns (C0-ECC-GFRP and C20-ECC-GFRP) were designed and fabricated. These columns were subjected to accelerated corrosion tests, where “C0” indicates no corrosion and “C20” represents a target corrosion rate of 20% for stirrups and longitudinal reinforcement. As shown in Figure 1, all columns had a cross-sectional diameter of 309 mm, a loading height of 1150 mm, and an aspect ratio of 3.72. To facilitate experimental loading, the columns were equipped with a loading end at the top, measuring 400 mm × 400 mm × 400 mm, and a base at the bottom, measuring 1400 mm × 600 mm × 400 mm. All columns were reinforced with six 20 mm-diameter HRB400 steel bars as longitudinal reinforcement, resulting in a longitudinal reinforcement ratio of 2.67%. The spiral reinforcement spacing in the region that was 400 mm from the column base was set to 40 mm, with the remaining spacing set to 80 mm. “T20” represents 20 mm-diameter HRB400E steel bars, and “D10” represents 10 mm-diameter HPB300 steel bars. The transverse reinforcement for ECC-GFRP spiral-confined RC columns utilized 10 mm-diameter GFRP spirals, whereas the traditional RC columns used 10 mm-diameter HPB300 steel spirals. Unlike traditional RC columns, which use HPB300 spiral and concrete protective layers, the protective layer of ECC-GFRP spiral-confined RC columns is made of prefabricated ECC-GFRP spirals tube to enhance corrosion resistance (Figure 1). The remaining longitudinal reinforcement of ECC-GFRP spiral-confined RC columns is consistent with that of traditional RC columns. To prevent a cold joint failure between ECCs and concrete at the column top and base during the experiment, the bond strength between the two materials was increased. The prefabricated ECC layer extended 100 mm into the column top and 300 mm into the base, with the embedded ECC surface roughened (Figure 1) [29].

### 2.2. Material Properties

This corrosion-resistant composite structural column incorporated multiple materials, including ECCs, concrete, GFRP spirals, steel spirals, and longitudinal steel rebar. The ECC mixture design followed a previous research by Li et al. [21], with detailed specifications listed in Table 1. The compressive and tensile strengths of the ECCs were obtained using standard testing methods. Based on ASTM C469 [37], uniaxial compression tests were conducted on three ECC cylindrical specimens with dimensions of Φ150 mm × 300 mm. The measured 28-day average compressive strength of the ECCs was 49.6 MPa. Additionally, uniaxial tensile tests were conducted on three dog-bone-shaped specimens, according to the standards of the Japan Society of Civil Engineers [38], to determine the tensile properties of the ECCs. Figure 2 shows the uniaxial compression and tensile stress–strain curves of the ECCs. The average tensile strength of all specimens was 7.4 MPa, and the ultimate tensile strain reached 3.8%. The concrete used in the ECCs and traditional reinforced-concrete columns was C40-grade commercial concrete. During the casting of the concrete columns, 4 sets of a total of 12 cubic specimens with dimensions of 100 mm × 100 mm × 100 mm were reserved. The measured 28-day average compressive strength of the concrete was 53.2 MPa. Additionally, uniaxial tensile tests were conducted on HRB400-grade longitudinal rebar and HPB300-grade spirals based on GB/T 228.1-2021 [39]. Figure 3 shows the uniaxial tensile stress–strain curves of HRB400 and HPB300 grade rebars. The elastic modulus of the HRB400 steel bar was 202 GPa with a yield strength of 438 MPa. The elastic modulus of the HPB300 steel bar was 205 GPa with a yield strength of 380 MPa. The mechanical parameters are summarized in Table 2. The material parameters of the GFRP spirals were taken from the manufacturer’s material test report, as shown in Table 3.

### 2.3. Specimen Fabrication

Unlike the conventional process for fabricating reinforced-concrete columns, the casting of ECC-GFRP spiral-confined RC columns involves two main stages: the fabrication of the ECC-GFRP tube and the casting of ordinary concrete. Figure 4 illustrates the step-by-step process of fabricating ECC-GFRP spiral-confined RC columns that was used in this work. (1) PVC pipes were used as the inner and outer formwork for fabricating the ECC-GFRP tube. The GFRP spiral was tied and fixed to the inner formwork according to the spacing designed in Figure 1. Then, the outer PVC formwork for the ECCs was installed and secured. In considering a ECC tube length of 1350 mm and a thickness of only 30 mm, an insertion-type vibrator could not be used for compacting (Figure 4a). Therefore, the ECC tube was cast and compacted by placing the PVC formwork horizontally on a vibrating table with side openings on the outer formwork to ensure proper compaction. (2) After 48 h of ECC casting, the inner and outer PVC formwork were removed, and then the ECC tubes were wrapped in wet geotextile and plastic film for 14 days of water curing (Figure 4b). (3) To enhance the bonding strength between the concrete and ECC interfaces, the ECC surface in contact with the concrete was roughened (Figure 4c). (4) The base rebar and longitudinal rebar of the column were made, and the prefabricated ECC-GFRP tube was placed in the design position as an outer formwork, i.e., fitted over the longitudinal rebar (Figure 4d). (5) Finally, the wooden formwork for the base and column cap were installed, and commercial concrete was cast (Figure 4e). (6) Two days after casting, the formwork was removed, and the ECC-GFRP RC columns were cured at ambient temperature for 28 days (Figure 4e).

### 2.4. Accelerated Corrosion Procedure

Considering that, during the serviceability life cycle of the structure, the plastic hinge region bears the main bending resistance under horizontal loads [40,41,42], and the bottom of the bridge columns is also located in the splash zone with severe environmental corrosion [43,44,45], the electrochemical accelerated corrosion area was set in this study within a 400 mm height from the bottom of the column, as shown in Figure 5. To accurately compare the corrosion resistance of the two types of columns, both the stirrups and longitudinal bars of the RC column were subjected to accelerated electric corrosion. Considering that, in real structures, the corrosion of stirrups is more severe, nylon tubes were used to insulate the stirrups and longitudinal bars to corrode them separately, with the target corrosion rate of the stirrups being 2.5% higher than that of the longitudinal bars [46]. Before casting the concrete, electric conduction wires were pre-embedded in the reinforcement cage, with the top of the reinforcement extending 200 mm for subsequent connections to the power supply (Figure 4f). As shown in Figure 5a, firstly, a stainless-steel mesh was wrapped around the corrosion area of the column, serving as the cathode, and then a PVC tube was placed on the outside, which was sealed at the bottom with glass glue and epoxy resin to prevent leakage; a NaCl solution with a salt concentration of 5% was added and left to stand for 3 days to ensure that the chloride ions penetrated the surface of the reinforcement. The longitudinal bars, stirrups, and stainless-steel mesh were connected to wires, with the longitudinal bars and stirrups connected to the positive pole of the power supply and the stainless-steel mesh connected to the negative pole. The corrosion current values were monitored by the readings of a DC ammeter, which is observed every 12 h. Additionally, according to Faraday’s law, the chemical change in the mass of the substance on the electrode interface is proportional to the electrical power passing through the material. The electrification time and the applied current to the specimen can be calculated using Equations (1) and (2), respectively:(1)$$t=\frac{\alpha Fr\;\rho }{Mi},$$
(2)I=inπdl,
where α represents the target corrosion level, F is the Faraday constant (96,500 C/mol), r is the radius of the rebar, ρ is the density of the rebar (7.85 g/cm^3^), M is the molar mass of iron (56 g/mol), and i is the current density (400 A/m^2^), n is the number of rebars, π represents the mathematical constant pi (approximated as 3.14), d is the diameter of the rebar, and l is the length of the corroded rebar. In this study, the designed corrosion rate for longitudinal rebars was 20%. Using Equations (1) and (2), the corresponding electrification time and current were calculated to be 1880 h and 0.603 A, respectively. For the steel spiral, the designed corrosion rate was 22.5%, resulting in an electrification time of 1880 h and a current of 0.599 A, as shown in Table 4. In order to ensure the stability of the applied current, the corrosion current value was monitored by the indicator meter of the DC meter, which was observed every 12 h.

### 2.5. Test Setup

To simulate the vertical loads and cyclic horizontal loads that concrete columns may encounter during earthquakes, the WAW-J12000 active follow-up loading system from Hangzhou Bangwei Electromechanical Control Engineering Co., Ltd. Hangzhou, China was used for testing, as shown in Figure 6. The top of the vertical loading actuator was connected to a rolling pulley, allowing the vertical actuator to follow the horizontal actuator when applying horizontal loads, ensuring that the axial force remains vertical throughout the loading process. A linear variable differential transformer (LVDT) was installed at the top of the column to measure the horizontal displacement at the top. Additionally, considering potential slippage at the base during horizontal load application, another LVDT was placed at the base to correct the top horizontal displacement (Figure 6b).

During the test, the axial load was applied using load control until the design value was reached and then held constant, with a loading rate of 50 kN/min. Horizontal loading was applied using displacement control, at a loading rate of 10 mm/min. Before applying horizontal loads, axial loading was performed first, applying 40% of the design vertical load (an axial compression ratio of 0.4). Then, a horizontal force that did not exceed 40% of the theoretical cracking load was applied in cyclic loading for three cycles. All bolts were then tightened again to eliminate the gaps between the fixtures and the specimen. During cyclic loading, the application of thrust was defined as positive loading, whereas the application of tensile force was defined as negative loading. All specimens were subjected to a push–pull loading sequence. The loading schedule, as shown in Figure 7, included the first four displacement levels of 2 mm, 4 mm, 8 mm, and 12 mm, each cycled once. Subsequently, the displacement increment was 8 mm, with each level cycled three times. The test was terminated when a sudden decrease in bearing capacity occurred due to core concrete crushing, rebar fracture, or GFRP spiral rupture. In this test, the side under compression during positive loading was defined as Face A, and during negative loading, it was defined as Face B (Figure 6b).

## 3. Test Observations

### 3.1. Evaluation of the Corrosion Condition

After the accelerated corrosion test, the corrosion patterns of the ECC-GFRP spiral-confined reinforced-concrete column and the conventional reinforced-concrete column were recorded, as shown in Figure 8. As seen in the figure, the corrosion products of the internal rebar exuded along the cracks on the surface of the columns, with the ECC-GFRP spiral-confined reinforced-concrete column (Figure 8a) exhibiting significantly fewer corrosion products than the traditional reinforced-concrete column (Figure 8b). The ECC material had an excellent crack control, and cracks are difficult to observe on the surface of the column (Figure 8a). This crack control ability further inhibited the penetration of corrosive agents under electrical conduction. Therefore, the C20-ECC-GFRP concrete column showed fewer corrosion products. For the traditional reinforced-concrete column (C20-RC in Figure 8b), the surface cracks were significant and accompanied by a substantial exudation of corrosion products, with some areas experiencing spalling of the concrete cover. The differences in the corrosion patterns of the two types of columns further demonstrated the excellent anti-corrosion capability of the ECC-GFRP spiral protective layer. It is noteworthy that the corrosion distribution along the circumference of the reinforced-concrete column was severely nonuniform. As shown in Figure 8b, Face A had significantly more corrosion products than Face B. This phenomenon also indicates the randomness of concrete cracking due to corrosion expansion, where the cracked areas further facilitate the transmission of corrosive agents, accelerating localized corrosion. However, the C20-ECC-GFRP concrete did not show significant localized corrosion due to the fine crack characteristics of ECC.

To more accurately assess the internal rebar corrosion condition, the corroded stirrups and longitudinal bars from the bottom 400 mm region of the columns were removed after the hysteresis test, and the actual corrosion rates were calculated. The rebar corrosion conditions for Specimens C20-RC and C20-ECC-GFRP are shown in Figure 9 and Figure 10, with their corresponding actual corrosion rates listed in Table 5. As seen in the figures, the longitudinal bars of C20-RC exhibited not only localized pitting, but also large cross-sectional corrosion (Figure 9a), with a corrosion rate of 14.81%. Notably, the stirrups showed severe corrosion (24.37%), with some sections nearly fractured after corrosion (Figure 9b). Thus, the corroded RC structure almost lost the confinement and reinforcement function of the stirrups. In contrast, the longitudinal bars of the C20-ECC-GFRP column mainly exhibited individual pitting, with its overall corrosion being significantly better than traditional reinforced concrete. Moreover, since the GFRP spiral bars replace steel, the C20-ECC-GFRP column had no stirrup corrosion. Compared to the traditional reinforced-concrete column, the actual longitudinal bar corrosion rate of C20-ECC-GFRP was only 11.59%, which is significantly lower than that of C20-RC. This further confirms that the ECC-GFRP spiral reinforcement protective layer effectively inhibits chloride ion penetration, exhibiting superior corrosion retardation capabilities for the rebar.

### 3.2. Failure Modes

The failure modes of ECC-GFRP spiral-confined reinforced-concrete columns and traditional reinforced-concrete columns under different corrosion levels are shown in Figure 11. The uncorroded reinforced-concrete column (C0-RC) exhibited a typical plastic hinge failure mode. In the initial stage, the crack features were mainly transverse bending cracks. As the displacement increased, the number of transverse bending cracks also increased, the concrete cover at the bottom of the column spalled, and the column ultimately failed due to concrete crushing (Figure 11a). For the corroded Specimen C20-RC, due to the expansive action of the corroded internal rebar, the specimen rapidly developed concentrated transverse cracks and vertical splitting cracks in the early stages of loading. Additionally, as the displacement increased, the concrete cover gradually spalled. The ultimate failure characteristic was the severe corrosion and subsequent fracture of the internal spiral rebar (Figure 11b). In contrast, due to the superior tensile properties of the ECC material, the crack characteristics of ECC-GFRP concrete showed numerous fine transverse cracks. The ECC-GFRP protective layer maintained good integrity even in the final stages of the specimen’s life. The final failure characteristic was the formation of a primary transverse bending crack at the column base and the concrete crushing, while the GFRP spiral remained intact, as shown in Figure 11c,d. Notably, for Specimen C0-ECC-GFRP, due to variations in the quality of the ECC layer during the manufacturing process, the column exhibited different failure modes under positive and negative loading—an unusual phenomenon. Under positive loading, the crack characteristics included the formation of a primary transverse crack at the column base, as predicted. However, under negative loading, the crack characteristics included concentrated vertical expansion cracks in the compression zone at the column base, displaying a shear failure mode (Figure 11c). This difference in failure phenomena further led to variations in the bearing capacity during the push–pull process. Detailed discussions are provided in the next section.

## 4. Results and Discussions

### 4.1. Hysteretic Responses

Figure 12 presents the hysteresis curves of the ECC-GFRP spiral-confined RC columns and traditional RC columns. Based on the hysteresis curves, the skeleton curves for each specimen are depicted in Figure 13. From the characteristics of the skeleton curves, three key points representing the mechanical performance of the RC columns were identified: yield point, peak point, and ultimate failure point (as shown in Figure 14). Since the load–displacement curves in this experiment did not show an obvious yield point, this study used the widely adopted equivalent energy method to calculate the yield point position [31,46]. As shown in Figure 14, a line parallel to the horizontal axis was drawn through the first peak Point B, and a line through Origin 0 intersected the parallel line and the skeleton curve at Points D and E, respectively, such that the areas of the two enclosed regions 0E and BDE were equal. A vertical line was then drawn through Point D to intersect the skeleton curve at Point A, which is referred to as the equivalent yield point of the skeleton curve [46]. The point where the curve descends to 85% of the peak load is referred to as the ultimate point [31]. Based on the above definitions, the key mechanical performance parameters obtained from the experimental load–displacement curves are listed in Table 6.

Comparing the uncorroded Specimen C0-RC with the corroded Specimen C20-RC revealed that the traditional reinforced-concrete specimen, after undergoing accelerated corrosion tests, exhibited a significantly pinched hysteresis loop due to severe corrosion of the stirrups. This indicates a substantial impact of corrosion on the seismic performance of reinforced concrete (Figure 12a,b). However, for the corroded Specimen C20-ECC-GFRP with ECC-GFRP used as a protective layer, the load–displacement hysteresis curve remained full despite undergoing severe corrosion. This demonstrates that the ECC-GFRP protective layer not only reduces the corrosion of the longitudinal bars, but also effectively prevents the reduction in confinement efficiency caused by the corrosion of the steel stirrups, thus ensuring the seismic performance of the concrete column in a corrosive environment. Notably, the C0-ECC-GFRP column exhibited significant differences in bearing capacity between positive and negative loading, which was attributed to the markedly different failure modes in the two directions (Figure 12c). This failure mode is atypical, so the data from this specimen are not included in subsequent analyses.

The skeleton curves (Figure 13) show that the negative bearing capacity of Specimen C20-RC is significantly lower than its positive bearing capacity, which is due to localized corrosion differences. As shown in Figure 8b of Section 3.1, the actual rebar corrosion on Face A was more severe than on Face B, causing stirrup fracture and longitudinal bar pullout during the negative loading on Face A, leading to a significant decrease in bearing capacity. In contrast, the positive loading curve showed a slightly higher bearing capacity than the C0-RC specimen due to the lower corrosion rate on Face B, with minor corrosion products filling the gaps between the concrete and rebar, enhancing the bond between the concrete and longitudinal bars. This study analyzed the seismic performance of Specimen C20-RC under the worst-case scenario, considering only the load–displacement curve under negative loading for comparison with other specimens. Combining Figure 13 and Table 6 for analysis, when compared to Specimen C0-RC, the peak bearing capacity of Specimen C20-RC under negative loading decreased by 51.65%, and the ultimate displacement decreased by 65.26%, indicating a significant reduction in the bearing capacity and ductility of the RC columns at a target corrosion rate of 20%. Compared to Specimen C20-RC, the peak bearing capacity of Specimen C20-ECC-GFRP under negative loading increased by 97.94%, and the ultimate displacement increased by 168.56%. This indicates that, at a rebar corrosion rate of 20%, the bearing capacity and deformation capacity of ECC-GFRP spiral-confined RC columns are far superior to those of traditional reinforced-concrete columns.

### 4.2. Seismic Performance of the ECC-GFRP Spiral-Confined RC Column

This section comprehensively evaluates the seismic performance of the ECC-GFRP spiral-confined reinforced-concrete columns, focusing on stiffness degradation, ductility, and energy dissipation.

The stiffness of RC columns plays a crucial role in limiting structural deformation during seismic events, minimizing horizontal displacement and sway amplitude, and thereby safeguarding the overall structural stability. This study determined the secant stiffness $K_{i}$ at each displacement level by connecting the peak points of the positive and negative loading directions on the skeleton curve, as illustrated in Figure 15. This methodology assesses the stiffness degradation of the specimens under low-cycle repeated loading [28] with the calculation formula presented in Equation (3).
(3)$$K_{i}=\frac{\left|+F_{i}\right|+\left|-F_{i}\right|}{\left|+\Delta_{i}\right|+\left|-\Delta_{i}\right|},$$
where $K_{i}$ represents the secant stiffness at the *i*-th displacement level and denotes the positive and negative peak loads at the *i*-th displacement level (red dotted line in Figure 15), and $+\Delta_{i}$ and $-\Delta_{i}$ indicate the positive and negative peak displacements at the *i*-th displacement level, respectively. The stiffness degradation derived from Equation (3) is illustrated in Figure 16. A comparison of the curves for Specimens C0-RC and C20-RC revealed that corrosion significantly impacted the initial stiffness of the RC specimens and markedly accelerated the rate of stiffness degradation. This is because the corrosion not only weakened the stiffness of the steel bars in the RC column (reduction in cross-sectional area), but it also weakened the bond between the steel bars and concrete. However, under identical corrosion conditions, the stiffness degradation trend of the ECC-GFRP-protected specimens (C20-ECC-GFRP) closely mirrored that of the uncorroded RC specimens (C0-RC), with both performing significantly better than the corroded RC specimens (C20-RC). At equivalent displacements, the stiffness of the C20-ECC-GFRP exceeded that of C20-RC. These experimental results indicate that the ECC-GFRP spiral-confined RC column can significantly mitigate stiffness degradation due to corrosion, thereby ensuring effective deformation control of the structure under extreme conditions.

The ductility of the RC columns refers to the structure’s capacity to undergo plastic deformation under load. During earthquakes, concrete columns with high ductility can maintain structural integrity even under substantial deformation, thereby preventing sudden failure. This study employed the displacement ductility coefficient to assess the ductility of the specimens [47]. The ductility coefficient $\mu_{\Delta }$ was determined by the ratio of the ultimate displacement to yield displacement derived from the skeleton curve, which given by Equation (4) as follows:(4)$$\mu_{\Delta }=\frac{\Delta_{u}}{\Delta_{y}},$$
where $\mu_{\Delta }$ denotes the displacement ductility coefficient; $\Delta_{y}$ signifies the yield displacement; and $\Delta_{u}$ represents the ultimate displacement. Using this formula, the ductility coefficients of the specimens were captured, as presented in Table 6 and Figure 17. As illustrated in this figure, the ductility coefficient of the corroded Specimen C20-RC (1.83) was markedly lower than that of the uncorroded Specimen C0-RC, indicating that rebar corrosion significantly diminishes the ductility of RC columns. However, under identical corrosion conditions, the ductility coefficient of Specimen C20-ECC-GFRP was 59.33% higher than that of Specimen C20-RC. This improvement is attributed to the ECC-GFRP spiral tube’s capability to delay internal rebar corrosion and its exceptional corrosion resistance, which prevent the premature stirrup failure caused by corrosion. The ECC-GFRP spiral tube continued to provide effective confinement to the concrete, thereby enhancing its ductility even after corrosion.

Structures with superior ductility can absorb and dissipate seismic energy through plastic deformation, thereby reducing structural damage. This study utilized cumulative energy dissipation as an indicator for quantitatively analyzing the energy dissipation capacity of the specimens by summing the areas of the hysteresis loops during the first cycle at each displacement level. For each specimen, the final cumulative energy dissipation point is defined as the horizontal displacement at which the load decreases to 85% of the peak load. Figure 18 illustrates the cumulative energy dissipation curves of each specimen. Compared to the uncorroded Specimen C0-RC, the final cumulative energy dissipation of the corroded column C20-RC decreased by 81%. However, under identical corrosion conditions, the final cumulative energy dissipation of Specimen C20-ECC-GFRP increased by 365% compared to that of Specimen C20-RC. This indicates that the energy dissipation capacity of ECC-GFRP spiral-confined RC columns in a corrosive environment is significantly superior to that of RC columns. The exceptional energy absorption and dissipation capabilities of the ECC-GFRP spiral-confined reinforced-concrete columns reduce structural acceleration and displacement amplitudes, thereby lowering stress levels in components, and effectively preventing local failures and overall collapse.

## 5. Finite Element Analysis

It is challenging to comprehensively evaluate the seismic performance of the novel ECC-GFRP spiral-confined RC columns in harsh environments based on limited test data. Consequently, this study employed numerical simulations to thoroughly investigate critical parameters, including longitudinal reinforcement ratio, shear span ratio, concrete compressive strength, and ECC compressive strength, to evaluate the seismic performance of the ECC-GFRP spiral-confined reinforced-concrete columns.

### 5.1. Finite Element Modeling

This study utilized the commercial finite element (FE) software ABAQUS 6.21 to model ECC-GFRP spiral-confined RC columns and analyze their seismic performance. The material parameters in the model are defined based on the experimental parameters in Section 2.2 or the data provided by the manufacturers. A bilinear hardening constitutive model was adopted for steel reinforcement (Figure 19a), and the constitutive relationship of the corroded steel followed the calculation method provided by Du et al. [48], where the effect of corrosion was considered by applying a reduction factor to the yield strength and tensile strength in the hardening stage, as illustrated in Figure 19a, with Equation (5) as follows:(5)$$\sigma_{y,cor}=\sigma_{y}\left(1-\alpha \eta \right),$$
where $\sigma_{y,cor}$ denotes the yield strength of the corroded rebar; α signifies the correction factor, taken as α=0.48; and η represents the corrosion rate of the steel. The tensile properties of GFRP employed a linear elastic constitutive model (Figure 19b). The ECC material adopted a simplified model based on the typical stress–strain curves obtained from uniaxial tension and compression tests [49]. The stress and strain characteristics of the material’s constitutive model were matched to the experimental data obtained in this study (Figure 2). Figure 19c,d demonstrate that the experimental data and the model exhibited a good match. The concrete material model adopted the improved FRP-confined concrete constitutive model proposed by Yu et al. [50] to account for the mechanical performance enhancement under confined stress, with the confinement efficiency defined and calculated through a user-defined subroutine. The material elements of the concrete and ECCs were modeled using C3D8R elements, while the material elements of the steel bar and GFRP spiral were modeled using T3D2 elements, with a model mesh size of 20 mm. To simplify the model and enhance computational efficiency, the longitudinal rebars and GFRP spiral were embedded into the matrix material. The bond–slip behavior between the rebars and concrete was simulated by incorporating spring elements. It was assumed that no relative displacement occurred between the rebars and concrete in the normal direction; therefore, the stiffness of the spring elements in the normal direction (the X and Y directions in this model) was set to be infinite. The bond–slip relationship between the rebars and concrete in the tangential direction (the Z direction in this model) was simulated by nonlinear spring elements, with the load–displacement behavior of the Z-direction spring elements following the bond–slip model for the ECC-GFRP spiral-confined concrete, as calculated by Expression (6).
(6)$$F=\tau \times \pi \times d\times l_{s},$$
where τ denotes the bond stress; *d* represents the diameter of the rebar; and $l_{s}$ signifies the distance between adjacent springs, which was taken as 20 mm in this model. The ECC extending into the column base was connected using the tie method, and all degrees of freedom of the column base were constrained. The loading method in the FE analysis was consistent with that described in Section 2.5, where a constant vertical load is applied followed by a monotonic horizontal displacement.

### 5.2. Model Verification

Figure 20 compares the skeleton curve results from the experiment (Specimen C20-ECC-GFRP) in this study and the model-predicted results from FE simulation. As shown in Figure 20, the model closely fit the ascending segment of the curve, while the descending segment of the model curve slightly deviated from the experimental curve. This deviation is attributed to the numerical model not accounting for the bond–slip issue between the ECCs and GFRP spiral. Therefore, the bond–slip relationship between stirrups and ECC was considered necessary. However, this issue is beyond the scope of the current study and will be addressed in future research. Despite this, the peak load obtained from the experiment was 128.2 kN, and the peak load from the model was 127.7 kN, with an error of only −0.38%, which further validates the effectiveness of the numerical simulation.

Figure 21 illustrates the comparison of the failure modes between the numerical simulation and the experiment, where the final failure mode of the model was the ECCs cracking at the column base and the concrete being crushed, matching the failure mode of Specimen C20-ECC-GFRP in the experiment. The results demonstrate that the established finite element model can effectively simulate the load–displacement behavior of ECC-GFRP spiral-confined RC columns.

### 5.3. Parameter Analysis

To further investigate the seismic performance of the ECC-GFRP spiral-confined RC columns under adverse environmental conditions, parameter analysis was supplemented based on the existing experimental research parameters by examining the influence of critical parameters, including the longitudinal reinforcement corrosion rate (*C_r_*), volumetric stirrup ratio (*ρ_sv_*), and the concrete compressive strength (*f_c_*). The reinforcement and dimensions of the control specimen for the FE parameter analysis were consistent with those of the experimental Specimen C20-ECC-GFRP. In addition, the stirrup spacing was altered to 40 mm throughout the entire length of the column, and the longitudinal reinforcement corrosion rate was set to 10%. The parameters of the control specimen and the range of values for each design parameter were presented, as shown in Table 7 and Table 8, respectively. The impact of each design parameter on the load–displacement curve of the specimens is illustrated in Figure 22, where the variation in volumetric stirrup ratio was achieved by altering the spiral spacing to 40 mm, 60 mm, and 80 mm.

The effects of corrosion rate *C_r_* on the load–displacement curve of the ECC-GFRP spiral-confined RC column is shown in Figure 22a. As observed from the figure, the load–displacement curve was not significantly affected by the corrosion rate before the peak. However, the descending curve for non-corrosion specimens (*Cr* = 0%) initially showed a rapidly descending trend in the post-peak region. However, as displacement increased, the trend changed with the increased corrosion rates. This is due to the fact that, for the column with *C_r_* = 0%, the tensile longitudinal bars did not yield at the peak point (Figure 23a), and the outermost ECC in the compression zone failed under compression, leading to a rapid decrease in post-peak bearing capacity. In contrast, the column with *C_r_* = 20% had a lower yield strength, causing the tensile longitudinal bars to yield before the peak point, thereby enhancing the column’s deformation capacity, as shown in Figure 23a. As displacement increased, the effectiveness of the GFRP spiral reinforcement in confining the core concrete became more pronounced. Additionally, in the non-corroded column, the tensile longitudinal bars yielded and exhibited a higher tensile strength compared to the corroded column, resulting in an increased flexural bearing capacity. Consequently, the lower the corrosion rate, the slower the curve descended, as shown in Figure 23b.

The influence of volumetric stirrup ratio *ρ_sv_* on the load–displacement curve of the ECC-GFRP spiral-confined RC column is shown in Figure 22b. As the volumetric stirrup ratio increased, the peak load capacity of the specimen showed a slight or negligible increase, and the post-peak curve descended more gently. This is because the elastic modulus of the GFRP spiral is relatively low, providing no significant confinement effect on the concrete before the peak load. After the peak, the confinement effect gradually increased, and increasing the number of stirrups enhanced the GFRP’s confinement effect, thereby improving the ultimate deformation capacity of the specimen.

The influence of concrete compressive strength *f_c_* on the load–displacement curve of the ECC-GFRP spiral-confined RC column is shown in Figure 22c. At the same displacement level, as the concrete compressive strength increased, the stiffness of the specimen increased and the peak load increased, but the post-peak load capacity declined more rapidly. This is because higher concrete compressive strength results in a higher elastic modulus, which increases the load-bearing capacity and stiffness of the column at the same displacement. However, since the volumetric stirrup ratio is the same for all specimens, the enhancement effect of confinement is greater for core concrete with a lower strength, resulting in a slower decline in the post-peak load capacity for lower-strength concrete.

### 5.4. Sensitivity Analysis

To investigate the sensitivity of various design parameters on the load-bearing capacity of ECC-GFRP spiral-confined RC columns under adverse environmental conditions, a control group specimen with a corrosion rate of 10%, a volumetric stirrup ratio of 3.02%, and a concrete compressive strength of 40 MPa was used. Sensitivity analysis was conducted on parameters including corrosion rate, volumetric stirrup ratio, and concrete compressive strength.

Table 9 and Figure 24 show the statistical table and sensitivity comparison chart of the impact of each parameter on the peak load capacity, respectively. It can be seen from this figure that changes in the volumetric stirrup ratio and concrete compressive strength exhibited a linear increasing trend with the peak load capacity, while the corrosion rate showed a negative correlation with the peak load capacity. Among these parameters, concrete compressive strength had the greatest impact on the peak load capacity of ECC-GFRP spiral-confined RC columns, while the corrosion rate had the least impact.

## 6. Conclusions

This study experimentally investigated the hysteresis behavior of corroded and non-corroded ECC-GFRP spiral-confined RC columns. It comprehensively analyzed key seismic parameters, including crack patterns, failure modes, hysteretic responses, load-bearing capacity, ductility performance, stiffness degradation, and energy dissipation capacity. Based on the experimental data, a finite element model was established to conduct parametric and sensitivity analyses, leading to the following main conclusions:(1)Compared to RC columns, ECC-GFRP spiral-confined RC columns exhibit a lower actual corrosion rate of the steel reinforcement, smaller crack widths after corrosion, and fewer corrosion products. The ECC-GFRP spiral tube demonstrates excellent crack control performance and corrosion resistance, effectively inhibiting steel reinforcement corrosion.(2)Hysteresis tests on corroded ECC-GFRP spiral-confined RC columns indicate that, at a target corrosion rate of 20%, these columns exhibit superior hysteretic performance. The load-bearing capacity, ductility, and energy dissipation capacity of ECC-GFRP spiral-confined RC columns are better than those of corroded RC columns, demonstrating excellent durability and seismic performance. The application of ECC-GFRP spiral-confined concrete reinforcement not only weakens the corrosion of the internal longitudinal reinforcement, but it can also provide stable confinement to the internal concrete, thereby ensuring the mechanical properties of the RC column in harsh environments.(3)Based on the experimental data, the accuracy of the finite element model was verified. The results of the parametric and sensitivity analyses indicate that, as the volumetric stirrup ratio and concrete compressive strength increase or the corrosion rate decreases, the peak load of the specimens improves. Among these factors, concrete compressive strength has the most significant impact on the load-bearing capacity of ECC-GFRP spiral-confined RC columns, while the corrosion rate has the least impact.

According to these test results and conclusions, some suggestions for future research directions on ECC-GFRP spiral-confined RC column structures are proposed: (1) the limitations of measuring instruments make it difficult to assess and compare the confinement effects between GFRP spirals and steel bars; (2) the bond–slip issue between ECC and GFRP spirals significantly affects the confinement effect of the ECC-GFRP spiral tube, requiring further research; and (3) the influence of variables such as the confinement stiffness and height of the ECC-GFRP spiral tube in the plastic hinge region on the seismic performance of the structure. 

Nevertheless, such composite structures still show great potential advantages in engineering applications. The ECC-GFRP spiral tube can be prefabricated as a concrete formwork in advance, reducing formwork engineering costs, significantly improving the construction quality and efficiency of the components, and providing good economic benefits. Additionally, this structure can significantly enhance the corrosion resistance and mechanical properties of RC structures, offering good practicality. Therefore, ECC-GFRP spiral-confined RC columns have broad engineering application prospects in coastal structures in earthquake-prone areas, such as bridge piers.

## Figures and Tables

**Figure 1 polymers-16-02110-f001:**
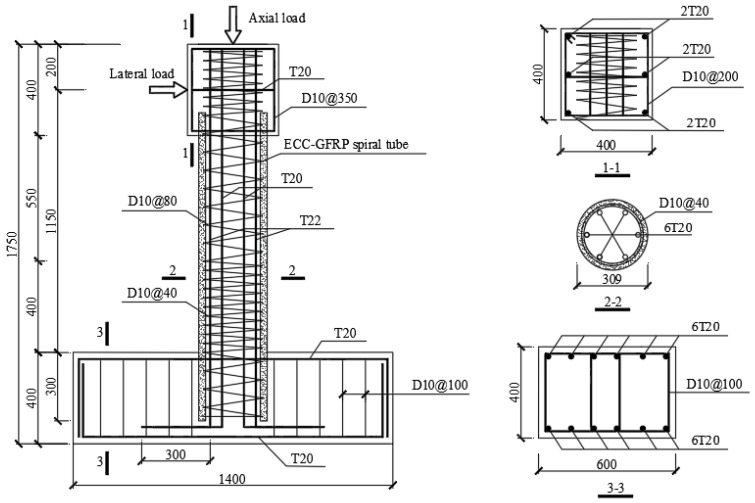
Specimen details of the ECC-GFRP spiral-confined RC column (unit: mm).

**Figure 2 polymers-16-02110-f002:**
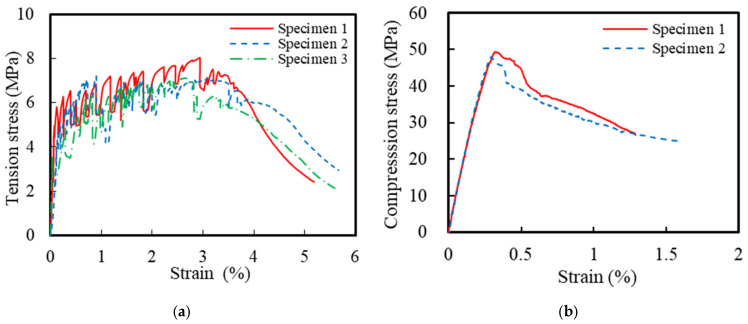
Stress–strain relationships of the ECCs: (**a**) tension and (**b**) compression.

**Figure 3 polymers-16-02110-f003:**
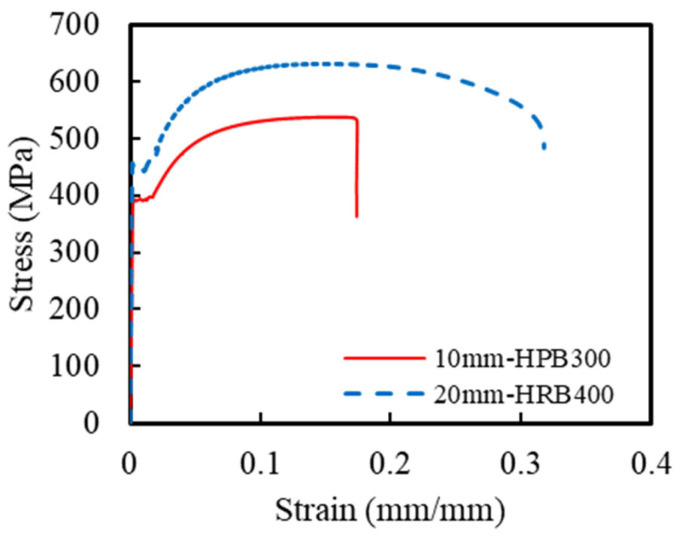
Stress–strain relationships of the steel.

**Figure 4 polymers-16-02110-f004:**
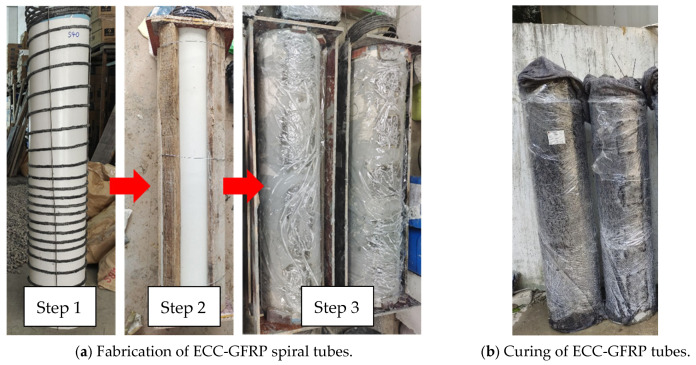
Fabrication of ECC-GFRP spiral-confined RC columns.

**Figure 5 polymers-16-02110-f005:**
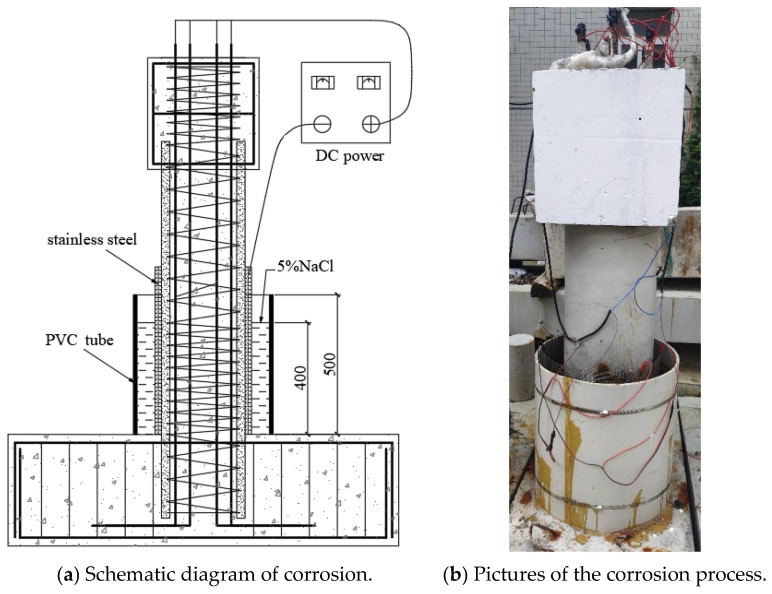
Electrochemically accelerated corrosion test.

**Figure 6 polymers-16-02110-f006:**
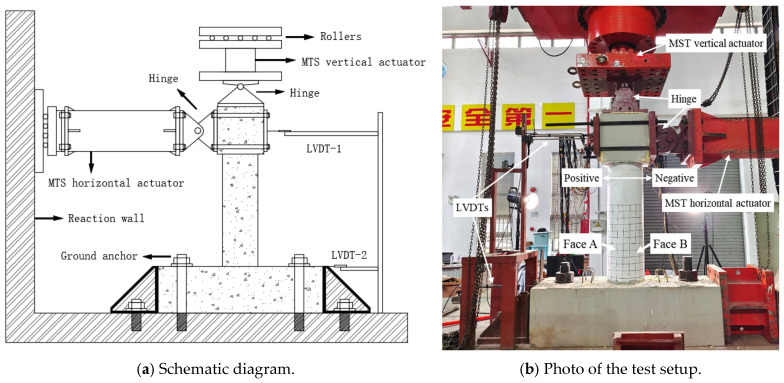
Test setup.

**Figure 7 polymers-16-02110-f007:**
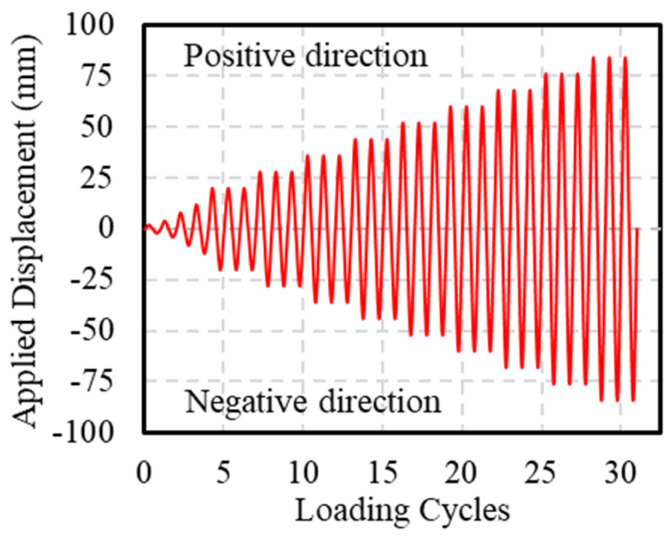
Cyclic loading plan.

**Figure 8 polymers-16-02110-f008:**
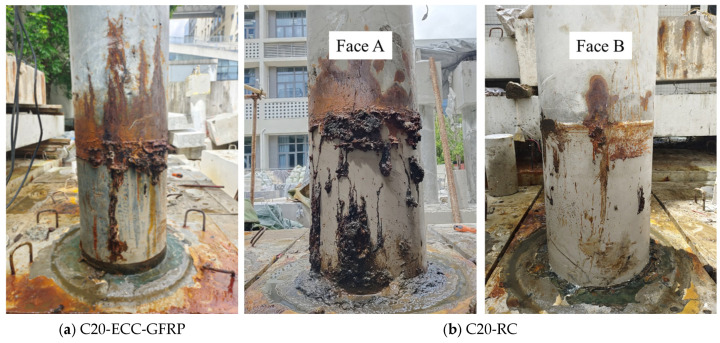
The corrosion status of specimens after accelerated corrosion.

**Figure 9 polymers-16-02110-f009:**
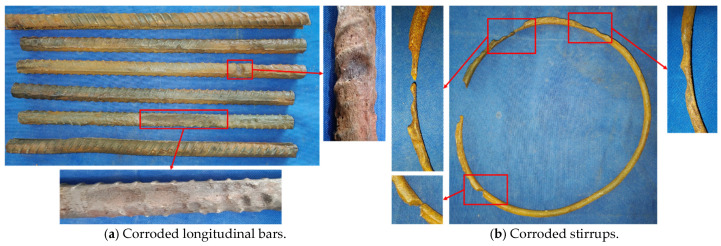
The corrosion condition of steel bars in C20-RC.

**Figure 10 polymers-16-02110-f010:**
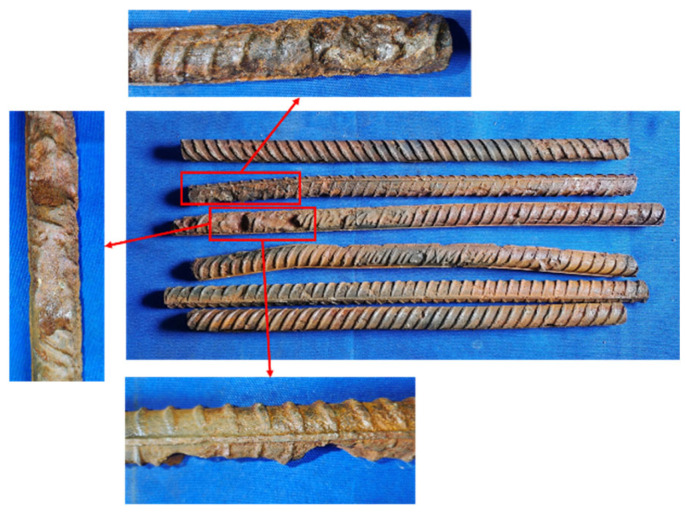
The corrosion condition of longitudinal bars in C20-ECC-GFRP.

**Figure 11 polymers-16-02110-f011:**
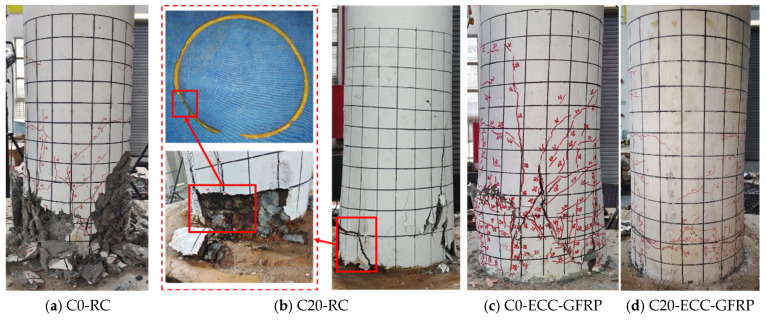
Failure modes.

**Figure 12 polymers-16-02110-f012:**
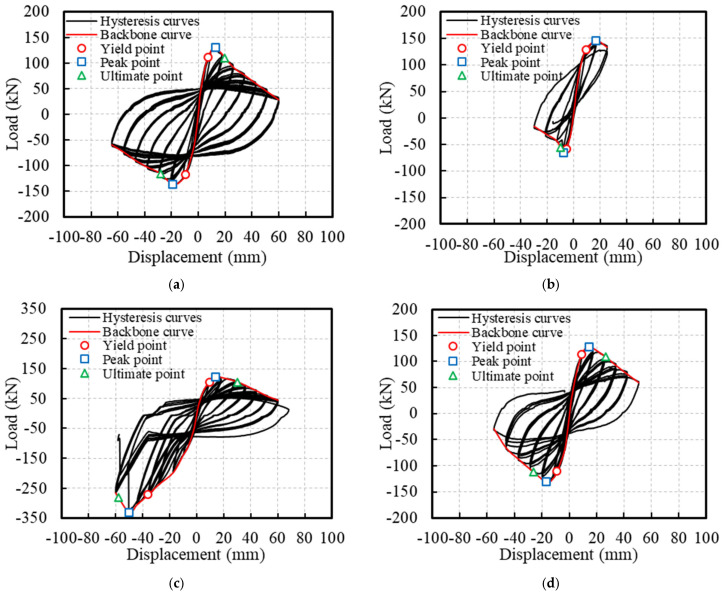
Hysteretic responses of the RC columns: (**a**) C0—RC; (**b**) C20—RC; (**c**) C0—ECC-GFRP; and (**d**) C20—ECC-GFRP.

**Figure 13 polymers-16-02110-f013:**
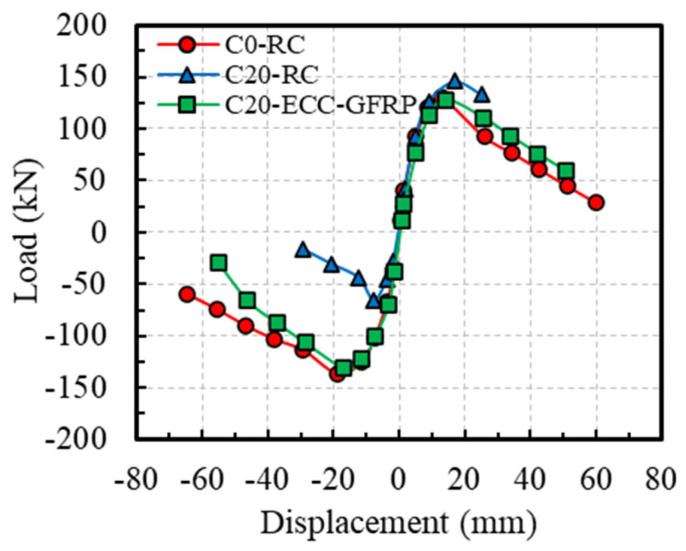
Backbone curves.

**Figure 14 polymers-16-02110-f014:**
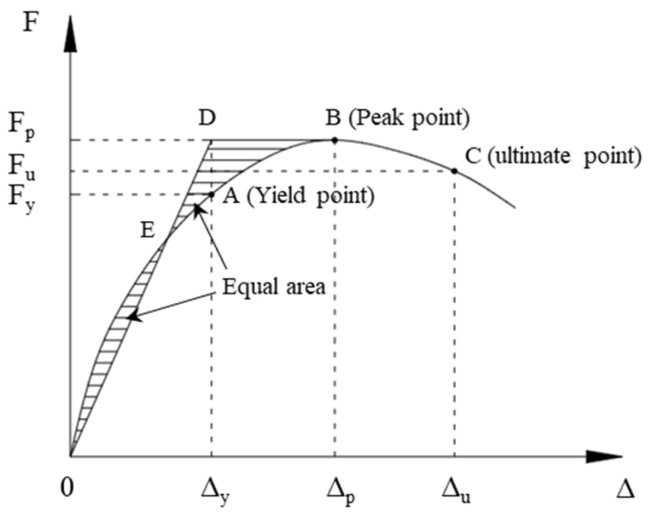
Equivalent energy method.

**Figure 15 polymers-16-02110-f015:**
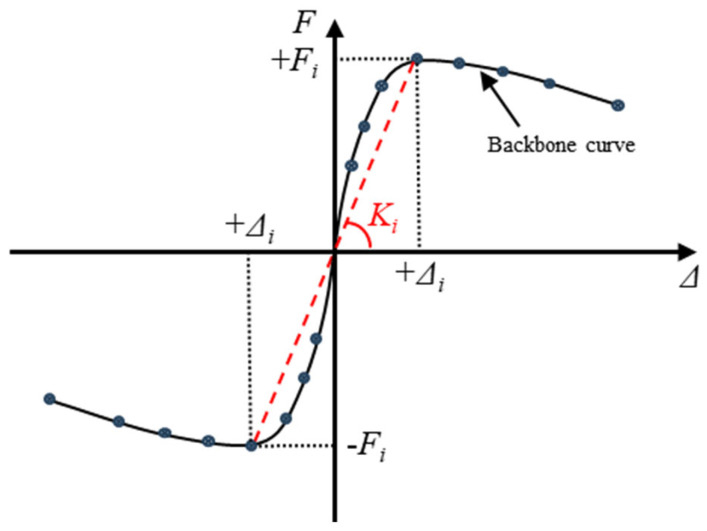
Stiffness calculation method.

**Figure 16 polymers-16-02110-f016:**
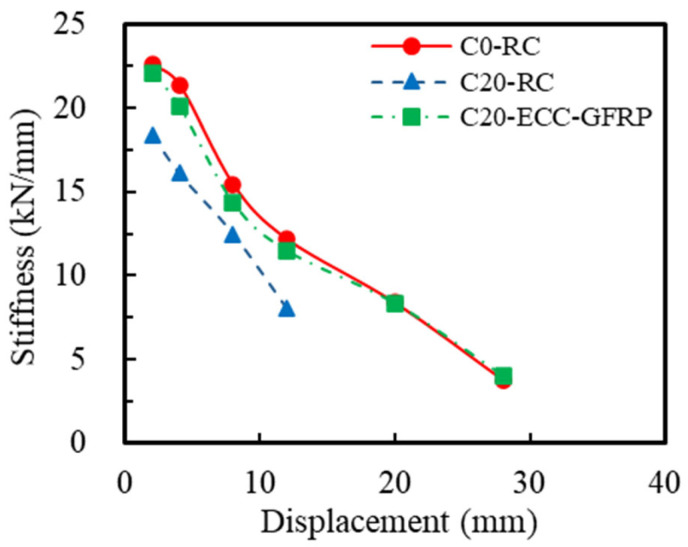
Stiffness deterioration.

**Figure 17 polymers-16-02110-f017:**
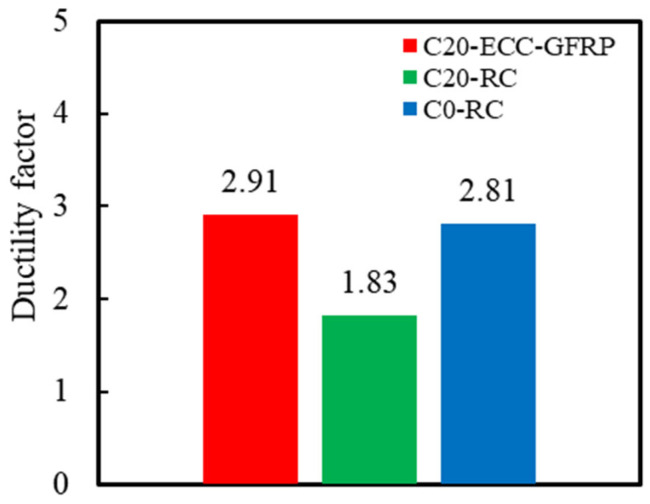
Ductility factor.

**Figure 18 polymers-16-02110-f018:**
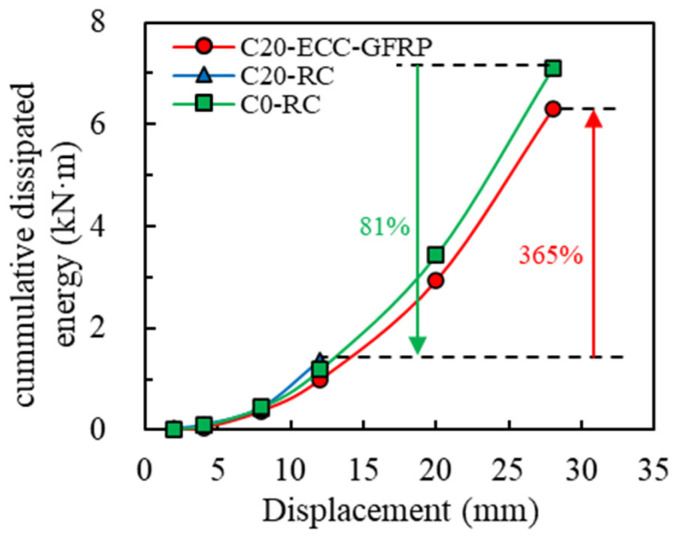
Accumulated energy dissipation curve.

**Figure 19 polymers-16-02110-f019:**
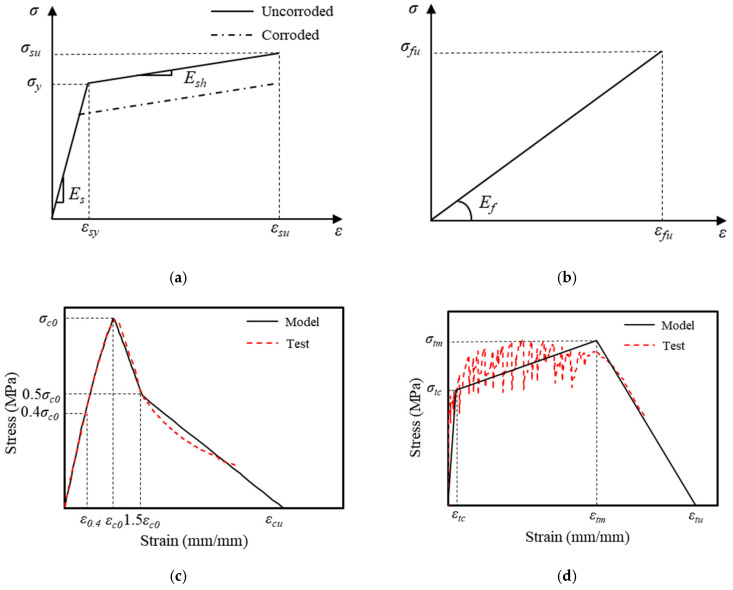
The constitutive model for the following materials: (**a**) steel; (**b**) GFRP; (**c**) ECC compression; and (**d**) ECC tension.

**Figure 20 polymers-16-02110-f020:**
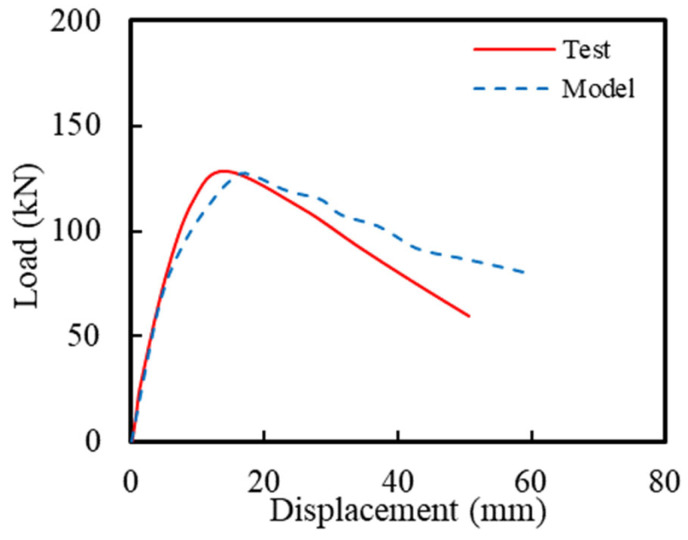
Comparison of the hysteretic curves.

**Figure 21 polymers-16-02110-f021:**
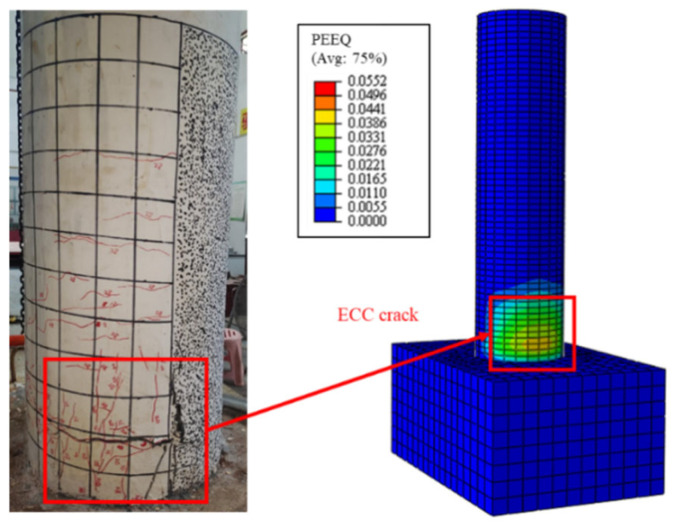
Comparison of the failure modes.

**Figure 22 polymers-16-02110-f022:**
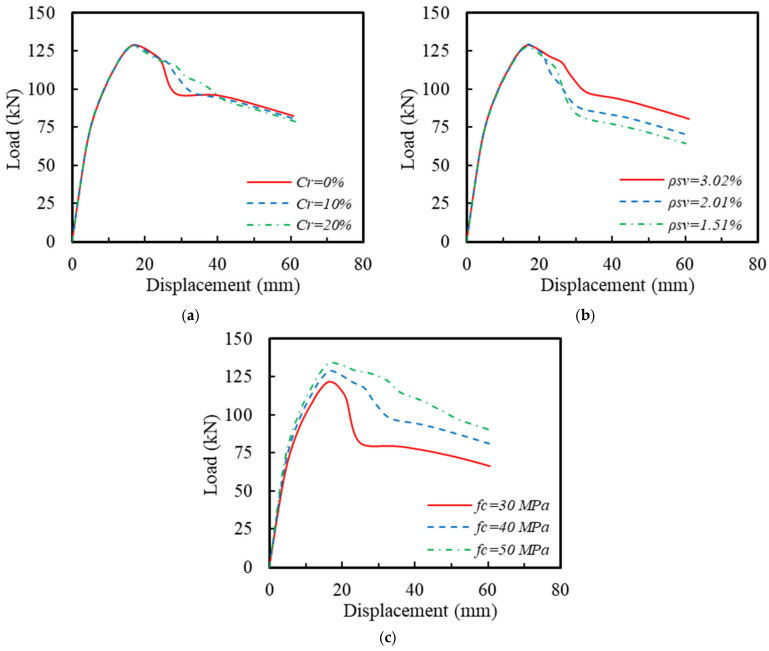
The influences of different parameters: (**a**) Corrosion rate *C_r_*; (**b**) Volumetric stirrup ratio *ρ_sv_*; (**c**) Concrete compressive strength *f_c_*.

**Figure 23 polymers-16-02110-f023:**
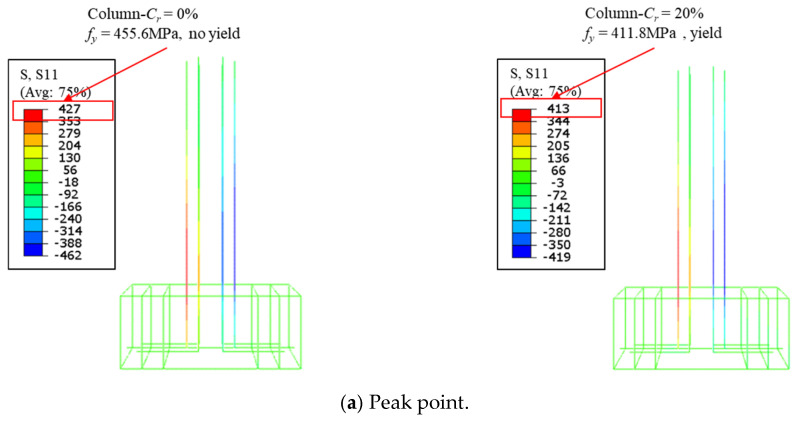
The effects of the corrosion rate on the mechanical performance of the columns.

**Figure 24 polymers-16-02110-f024:**
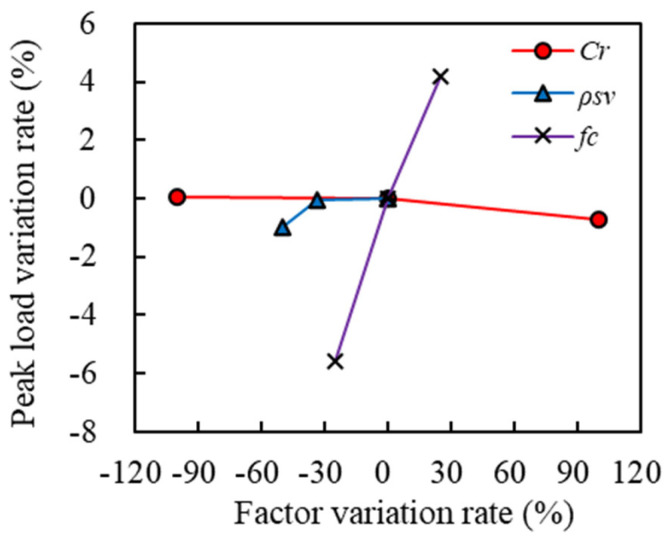
Sensitivity study.

**Table 1 polymers-16-02110-t001:** Mix proportion of the ECCs.

Ingredients	Quantities
Limestone powder (LP) (kg/m^3^)	74
Silica fume (SF) (kg/m^3^)	111
Ground granulated blast furnace slag (GGBFS) (kg/m^3^)	566
Silica sand (kg/m^3^)	466
Cement (kg/m^3^)	528
Water (kg/m^3^)	388
HRWR (kg/m^3^)	3
PE fiber (kg/m^3^)	20
W*/(Cement + LP + SF + GGBFS)	0.303

**Table 2 polymers-16-02110-t002:** Mechanical properties of the steel.

Reinforcement Type	*d_s_* (mm)	*f_y_* (MPa)	*ε_y_* (mm/mm)	*f_su_* (MPa)	*ε_su_* (mm/mm)	*E_s_* (GPa)
Spiral bar	10 mm	380	0.002065	536	0.1440	205
Longitudinal bar	20 mm	438	0.002086	617	0.1510	202

**Table 3 polymers-16-02110-t003:** Mechanical properties of the GFRP.

Reinforcement Type	Diameter (mm)	Elasticity Modulus (GPa)	Ultimate Strain (%)	Ultimate Strength (MPa)
GFRP	10	46.8	2.03	791

**Table 4 polymers-16-02110-t004:** Calculation of the current magnitude and duration of the applied current.

Specimens	Reinforcement Type	Corrosion Rate (%)	Current Density (mA/mm^2^)	Current Value (A)	Time (h)
C20-ECC-GFRP	Longitudinal bar	20	0.4	0.603	1880
C20-RC	Longitudinal bar	20	0.4	0.603	1880
	Spiral bar	22.5	0.4	0.599	1880

**Table 5 polymers-16-02110-t005:** The actual corrosion rate of steel reinforcement.

Specimens	Reinforcement Type	Target Corrosion Rate	Actual Corrosion Rate
C20-ECC-GFRP	Longitudinal bar	20%	11.59%
C20-RC	Longitudinal bar	20%	14.81%
	Spiral bar	22.50%	24.37%

**Table 6 polymers-16-02110-t006:** Ductility factor.

Specimen	Direction	Yield Point		Peak Point		Ultimate Point		$\mu_{\Delta }$
		$F_{y}$ (kN)	$\Delta_{y}$ (mm)	$F_{p}$ (kN)	$\Delta_{p}$ (mm)	$F_{u}$ (kN)	$\Delta_{u}$ (mm)	
C20-ECC-GFRP	Positive	113.45	9.25	128.20	2.85	108.97	26.34	-
	Negative	109.83	−8.85	−130.54	2.97	−110.96	−26.31	-
	Average	111.64	9.05	-	2.91	109.96	26.33	2.91
C0-RC	Positive	110.87	7.22	130.28	2.73	110.74	19.68	-
	Negative	117.48	−9.82	−136.41	2.87	115.95	−28.20	-
	Average	114.18	8.52	-	2.81	113.34	23.94	2.81
C20-RC	Negative	−57.83	−5.37	−65.95	1.83	−56.06	−9.80	1.83

**Table 7 polymers-16-02110-t007:** Details of the control specimen.

*μ*	*ρ_sv_* (%)	*ρ_s_* (%)	*L*/*d*	*f_c_* (MPa)	*C_r_* (%)
0.4	3.02%	2.67	4	40	10

**Table 8 polymers-16-02110-t008:** Parameter analysis.

Design Factor	Range of Parameter Values
Longitudinal reinforcement corrosion rate *C_r_* (%)	0, 10, 20
Volumetric stirrup ratio *ρ_sv_* (mm)	40, 60, 80
Concrete compressive strength *f_c_* (MPa)	30, 40, 50

**Table 9 polymers-16-02110-t009:** Sensitivity study.

Specimen	*C_r_* (%)	*ρ_sv_* (MPa)	*f_c_* (MPa)	*F_p_* (kN)	*F_p_*/*F_p,control_*
Control specimen	10%	3.02%	40	129.14	100.00%
C0-S40-C40	0%			129.18	100.03%
C20-S40-C40	20%			128.18	99.26%
C10-S60-C40		2.01%		129.08	99.96%
C10-S80-C40		1.51%		127.86	99.01%
C10-S40-C30			30	121.93	94.42%
C10-S40-C50			50	134.55	104.19%

## Data Availability

Data are contained within the article.

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
