# Peer review of "Seismic Performance of Corroded ECC-GFRP Spiral-Confined Reinforced-Concrete Column"

_polymers, 2024, doi:10.3390/polym16152110_

Round 1
Reviewer 1 Report
Comments and Suggestions for Authors
Reviewer's comments:
Before publication, the authors must correct the mistakes in the text that are found all over the manuscript and address the following:
1. The authors can brief that how were the target corrosion rates achieved and controlled during the experiments?
2. Could the authors elaborate on the methods used to measure crack widths and corrosion products in ECC-GFRP spiral confined RC columns?
3. The authors need to be explain, what particular parameters were included in the finite element model, and in what manner were they validated against experimental data?
4. Could the authors provide more details on the hysteretic responses they saw or observed in the corroded ECC-GFRP spiral confined RC columns?
5. In what ways did the tests' stiffness degradation between the corroded and non-corroded columns differ?
6. Can authors discuss the relative effects of corrosion rate versus other factors or parameters on the peak load and overall performance?
7. How do you propose to address the issue of bond slip between ECC and GFRP spiral in future studies?
8. Can the authors estimate the potential cost savings in formwork engineering and construction efficiency when using ECC-GFRP spiral tubes?
9. The authors should be discuss in conclusion section what are the key factors that contribute to the improved corrosion resistance and mechanical properties of ECC-GFRP spiral confined RC columns?
10. The author should need to add at least 3-5 new references from anywhere or specially the "Polymers" relevant to his topics.
11. Overall, the work is interesting; it just needs to follow the suggestions to improve the manuscript.
Comments on the Quality of English LanguageReviewer's comments:
Before publication, the authors must correct the mistakes in the text that are found all over the manuscript and address the following:
1. The authors can brief that how were the target corrosion rates achieved and controlled during the experiments?
2. Could the authors elaborate on the methods used to measure crack widths and corrosion products in ECC-GFRP spiral confined RC columns?
3. The authors need to be explain, what particular parameters were included in the finite element model, and in what manner were they validated against experimental data?
4. Could the authors provide more details on the hysteretic responses they saw or observed in the corroded ECC-GFRP spiral confined RC columns?
5. In what ways did the tests' stiffness degradation between the corroded and non-corroded columns differ?
6. Can authors discuss the relative effects of corrosion rate versus other factors or parameters on the peak load and overall performance?
7. How do you propose to address the issue of bond slip between ECC and GFRP spiral in future studies?
8. Can the authors estimate the potential cost savings in formwork engineering and construction efficiency when using ECC-GFRP spiral tubes?
9. The authors should be discuss in conclusion section what are the key factors that contribute to the improved corrosion resistance and mechanical properties of ECC-GFRP spiral confined RC columns?
10. The author should need to add at least 3-5 new references from anywhere or specially the "Polymers" relevant to his topics.
11. Overall, the work is interesting; it just needs to follow the suggestions to improve the manuscript.
Author Response
Comment 1. The authors can brief that how were the target corrosion rates achieved and controlled during the experiments?
Response 1:
According to Faraday's law, the chemical change in the mass of the substance on the elec-trode interface is proportional to the electrical power passing through the material. Therefore, known the target corrosion rates, the electrification time and the applied current to the specimen can be calculated using equations (1) and (2). Meanwhile, the corrosion current value is monitored by the indicator meter of the DC meter, which is observed every 12 hours.
The related explanation has been added to the revised manuscript: Line 244-246.
Comment 2. Could the authors elaborate on the methods used to measure crack widths and corrosion products in ECC-GFRP spiral confined RC columns?
Response 2:
We thank the reviewer for raising this question. In this study, the crack width was measured by a mobile crack width measuring instrument during tests. As shown in Fig. 11, ECC-GFRP columns show a significant difference compared to traditional RC columns. This study describes the development trend of cracks and the shape of cracks during the loading process (Figure 11) and compares the surface crack pattern of the two corrosion columns through observation (Section 3.1 and 3.2). The comparison of this observation method has significantly proved that ECC-GFRP Colum has extremely excellent crack control capabilities. Therefore, quantifying and comparing crack widths is not of particular significance in such cases, which the reason why the data of crack width did not provide.
Comment 3. The authors need to explain what particular parameters were included in the finite element model, and in what manner were they validated against experimental data?
Response 3:
First, the FE model considers the effect of rebar corrosion by applying a reduction factor to the yield strength and tensile strength, and the calculated method was given in Eq.5.
Second, the ECC material adopted a simplified model based on typical stress-strain curves ob-tained from uniaxial tension and compression tests which is different from conventional concrete materials.
Third, the concrete material model adopts the improved confined concrete constitutive model that considers the ECC and FRP spiral confinement effects.
Fourth, the bond-slip behavior between rebars and concrete is simulated by incorporating spring elements.
All the above special parameter settings are all to be closer to the actual test situation. Figures 20 compares the skeleton curve results from the experiment (specimen C20-ECC-GFRP) in this study and the model predicted results from FE simulation, which illustrates that the model closely fits the ascending segment of the curve.
The related explanation and discussion have been added to Section 5.1 and 5.2.
Comment 4. Could the authors provide more details on the hysteretic responses they saw or observed in the corroded ECC-GFRP spiral confined RC columns?
Response 4:
Thanks for the suggestions, and the detailed discussion has been introduced in Section 4.1.
For the corroded specimen C20-ECC-GFRP with ECC-GFRP used as a protective layer, the load-displacement hysteresis curve remained full despite undergoing severe corrosion. This demonstrates that the ECC-GFRP protective layer not only reduced the corrosion of the longitudinal bars but also effectively prevented the reduction in confinement efficiency caused by the corrosion of steel stirrups, thus ensuring the seismic performance of the concrete column in a corrosive environment.
Comment 5. In what ways did the tests' stiffness degradation between the corroded and non-corroded columns differ?
Response 5:
This is because corrosion not only weakens the stiffness of the steel bars in the RC column (reduction in cross-sectional area), but also weakens the bond between the steel bars and concrete. Therefore, as shown in Figure 16, the corroded RC column exhibits a significantly smaller initial stiffness and markedly accelerates the rate of stiffness degradation. However, under identical corrosion conditions, the stiffness degradation trend of ECC-GFRP protected specimens (C20-ECC-GFRP) closely mirrors that of uncorroded RC specimens (C0-RC), with both performing significantly better than the corroded RC specimens (C20-RC).
These related discussions have been added to the revised manuscript, Line 422-427.
Comment 6. Can authors discuss the relative effects of corrosion rate versus other factors or parameters on the peak load and overall performance?
Response 6:
The discussion about he corrosion rate on the peak load has been provided in Section 4.1, Line 381-400, and Table 6.
Comment 7. How do you propose to address the issue of bond slip between ECC and GFRP spiral in future studies?
Response 7:
This is a very good question. It is very challenging to consider the slip between ECC and GFRP. We plan to measure the strain of GFRP by embedding strain gauges, and meanwhile measure the ECC surface strain at the corresponding position while applying tension load. In this way, we can study the slip relationship between ECC and GFRP by analyzing their strain difference. This is only a preliminary plan at present, and we will refine this experimental method in subsequent work.
Comment 8. Can the authors estimate the potential cost savings in formwork engineering and construction efficiency when using ECC-GFRP spiral tubes?
Response 8:
We agree with the author's suggestion. However, the cost analysis requires systematic consideration of factors such as materials, design methods, construction costs, and long-term maintenance. Due to page limitations, this content is beyond the scope of this article. However, we can roughly estimate that this ECC-GFRP confined concrete column can save the cost of RC columns in harsh environments. The use of prefabricated ECC-GFRP formwork can save labor costs, and the reduction in long-term maintenance costs can also greatly reduce the maintenance costs of the column's service life. Therefore, this type of ECC-GFRP confined concrete column can achieve economic benefits.
Comment 9. The authors should be discuss in conclusion section what are the key factors that contribute to the improved corrosion resistance and mechanical properties of ECC-GFRP spiral confined RC columns?
Response 9:
In accordance with the reviewer's requirements, the conclusion has been revised.
(1) Compared to RC columns, ECC-GFRP spiral confined RC columns exhibit a lower actual corrosion rate of the steel reinforcement, smaller crack widths after corrosion, and fewer corrosion products. The ECC-GFRP spiral tube demonstrates excellent crack control performance and corrosion resistance, effectively inhibiting steel reinforcement corrosion.
(2) Hysteresis tests on corroded ECC-GFRP spiral confined RC columns indicate that at a target corrosion rate of 20%, these columns exhibit superior hysteretic performance. The load-bearing capacity, ductility, and energy dissipation capacity of ECC-GFRP spiral confined RC columns are better than those of corroded RC columns, demonstrating excellent durability and seismic performance. The application of ECC-GFRP spiral reinforcement not only weakens the corrosion of the internal longitudinal reinforcement, but also can provide stable confinement to the internal concrete, thereby ensuring the mechanical properties of the RC column in harsh environments.
Comment 10. The author should need to add at least 3-5 new references from anywhere or specially the "Polymers" relevant to his topics.
Response 10:
A key innovation of this work is to use GFRP as the circumferential restraint material to avoid corrosion of the restraint steel bars. FRP (Fiber reinforced polymer) is a typical POLYMER material, and related research has been cited many times in the introduction. In accordance with the reviewer's requirements, we expanded the application scenarios of FRP and added relevant literature citations.
Comment 11. Overall, the work is interesting; it just needs to follow the suggestions to improve the manuscript.
Response 11:
Many thanks to the reviewers for their recognition of our work
Reviewer 2 Report
Comments and Suggestions for Authors
Dear authors
You can find my remarks in the PDF file

Author Response
General comments:
The paper studies the seismic performance of corroded ECC-GFRP spiral confined reinforced concrete columns. The study combines an illustrative experimental part and a strong mathematical background.
The paper is very well prepared: the abstract is a complete summary, the paper is well organized, and the conclusions are clear. The presentation is very clear to people who are not experts in the subject. I have a few minor remarks for correction and improvement of the manuscript.
General Response:
Many thanks to the reviewers for their recognition of our work
Comments 1:
Table 1:Please add the abbreviations in the corresponding lines:
Limestone Powder (LP)
Silica fume (SF)
Ground granulated blast furnace slag (GGBFS)
Response 2:
As the reviewer commented, the abbreviations have been added to the revised manuscript.
Comments 2:
Please specify if GGBFS and silica sand are dry or humid.
If the materials are dry, then W*/(LP+SF+GGBFS+Cement)=0.303
If the two mentioned materials are humid, then W* is the sum of the fresh water and the
kg/m3 of humidity.
Please clarify the above.
Response 2:
Many thanks to the reviewer for pointing out this mistake. The GGBFS and silica sand are dry, and the calculated W*/(LP+SF+GGBFS+Cement) has been corrected to 0.303.
Round 2
Reviewer 1 Report
Comments and Suggestions for Authors
Reviewer' comments:
Reviewer #: In the present work (Revised Manuscript MDPI-polymers-3105402 V2) "
"After a thorough review of the author's revised manuscript (Manuscript MDPI-polymers-3105402 V2), I conclude that they have provided his answers to the reviewers' suggested comments with appropriate explanations.
Therefore, I accept and recommend this version related to the Polymers."